# BOUNDING THE ROBUSTNESS AND GENERALIZATION FOR INDIVIDUAL TREATMENT EFFECT

## ABSTRACT

Individual treatment effect (ITE) estimation has important applications in fields such as healthcare, economics and education, hence attracted increasing attention from both research and industrial community. However, most existing models may not perform well in practice due to the lack of robustness of the ITE estimation predicted by deep neural networks when an imperceptible perturbation has been added to the covariate. To alleviate this problem, in this paper, we first derive an informative generalization bound that demonstrate the expected ITE estimation error is bounded by one of the most important term, the Lipschitz constant of ITE model. In addition, in order to use Integral Probability Metrics (IPM) to measure distances between distributions, we also obtain explicit bounds for the Wasserstein (WASS) and Maximum Mean Discrepancy (MMD) distances. More specifically, we propose two types of regularizations called Lipschitz Regularization and reproducing kernel Hilbert space (RKHS) Regularization for encouraging robustness in estimating ITE from observational data. To the best of our knowledge, this is the first work on robustness for ITE estimation. Extensive experiments on both synthetic examples and standard benchmarks demonstrate our framework's effectiveness and generality. To benefit this research direction, we release our project at https://github-rite.github.io/rite/.

## 1 INTRODUCTION

Understanding the Individual Treatment Effect (ITE) of an treatment $T$ (e.g., a plan of drug) on an individual with features $X$ (e.g., demographic characteristics ) is of great importance across many domains, such as healthcare Shalit (2020), computer vision Santurkar et al. (2019); Elsayed et al. (2018) and recommender system Wang et al. (2021; 2022). It basically aims to discover the underlying patterns of the outcome $Y$ (e.g., patient's blood pressure) when receiving a specific treatment plan. Ideally, practitioners can measure the ITE based on randomized controlled trials (RCTs) in which the treatment assignment is randomized, and thus it is independent of the individual's features. However, RCTs are often a time-consuming process, and sometimes they are even unethical or illegal Qin et al. (2021). To solve this issue, both academic researchers and industrial practitioners tend to use easily accessed and available observational data for doing ITE estimation. The observational data usually consist of unit's covariates, treatments and outcomes. In the binary treatment case, the group of units receiving the treatment is called treated group, and others the control group. Due to the fact that the generating process of observational data is not under control, there exists an imbalanced distributions between treated and control groups, which hinders the estimation of ITE accurately and correctly. In the past few years, quite a lot of promising ITE models have been proposed and achieved impressive performance. For example, the representative CFR Shalit et al. (2017) method enforce the similarity between the distributions of treated and control groups in the representation space by a penalty term IPM.

In observational studies, however, robustness model for ITE is of great importance. Indeed when presented with a malicious individual's features that consist of an imperceptible perturbation to model, they can predict incorrect treatment effects with high-confidence and further make a wrong decision. For example, in one scenario where an imperceptible and malicious error is randomly added to the Electrocardiogram (ECG) data for the heart patients. Accroding to that results, doctors could make a wrong treatment plans, which would lead to disastrous consequences. Additionally, when we attempt to balance the distribution between treated and control groups, the lack of robustness

hinders the applications of MMD and WASS to achieve the optimal distribution alignment. To alleviate this problems, in this paper, we propose a novel and effective framework to achieve **R**obust **I**ndividual **T**reatment **E**ffect estimation (called RITE for short). The RITE framework enhence the robustness of ITE models, improving the generalization performance of ITE estimation. Driven by the aforementioned examples, we first derive an informative generalization error bound of the expected Precision in Estimation of Heterogeneous Effect (PEHE) loss, which demonstrate the expected ITE estimation error is bounded by one of the most important term, the Lipschitz constant of ITE model. And then, in order to apply IPM to balance the distributions between treated and control groups, we also obtain explicit generalization bounds for the WASS and MMD distances. The theoretical analysis clearly indicate that taking into account the Lipschitz constant of ITE model, along with empirical factual losses and the discrepancy between treated and control groups, we can greatly reduce the bounds of target loss function. Based on the above theory, we proposed two types of regularization called Lipschitz regularization and RKHS regularization respectively. The former aims to reduce the overall Lipschitz constant of ITE model by constraining its parameters towards the conditions of orthonormality, such that we can make WASS adaptive to the upper bound of objective . More concretely, we sum all layers constrain and add it to the target loss and by a hyperparameter to control its weight. While the latter is mainly focused on the constraint of products on RKHS, such that we can instantiate MMD in our settings. Analogous to Lipschitz regularization, we also add it to the target loss. The proposed two regularization are both to make IPM metric effective for encouraging robustness in estimating ITE from observational data.

In a summary, the main contributions of this paper can be concluded as follows: (1) We examine the problem of robustness in estimating ITE, deriving an informative generalization error bound for the PEHE loss. The derived bound can connect the robustness with adversarial machine learning and it is instructive for reducing the Lipschitz constant of ITE model. To the best of our knowledge, this is the first work on robustness for ITE estimation; (2) We obtain explicit bounds for the WASS and MMD distances; (3) According to the theoretical analysis, we propose a computationally efficient framework for encouraging robustness in estimating ITE from observational data; (4) We conduct extensive experiments based on both synthetic examples and standard benchmark datasets to validate its effectiveness.

## 2 PROBLEM FORMULATION

### 2.1 PROBLEM SETUP

We formulize our problems using the Neyman-Rubin potential outcomes framework Rubin (2005), as follows. Let $x \in \mathcal{X}$ denotes a unit features or covariates, $t \in \mathcal{T}$ stands for a treatment or intervention on a unit. Throughout this paper, we focus on the binary treatment case, where $\mathcal{T} = \{0, 1\}$ and $y \in \mathcal{Y}$ represents the factual outcome. In practice, we can only observe the factual outcome with respect to treatment assignment, i.e., $y = Y_0$ if $t = 0$, otherwise $y = Y_1$, where $Y_t$ denotes the potential outcome for treatment $t$. The Individual Treatment Effect (ITE) on a unit $x$, or also known as the conditional average treatment effect (CATE) Shalit et al. (2017):

$$\tau(x) := \mathbb{E}[Y_1 - Y_0 | x] \tag{1}$$

The fundamental problem of causal inference is that for any unit $x$ in our settings we only observe $Y_1$ or $Y_0$, but never both. Following recent works Qin et al. (2021); Wager & Athey (2018), we implicitly assume that there exists plenty of observable data. Formally, Let $D = \{(x_i, t_i, y_i)\}_{i=1}^m$ denote the training data drawn from the obserational data distribution $\mathcal{D}$, i.e. $D \in \mathcal{D}$. In order to guarantee that the potential outcomes are identifiable from factual observational data, the four assumptions are required: **Stable Unit Treatment Value Assumption(SUTVA), Consistency, Ignorability** and **Positivity** Yao et al. (2021). Based on above assumptions, we can formulate the problem of estimating ITE as: $\tau(x) = \mathbb{E}[Y|x, t = 1] - \mathbb{E}[Y|x, t = 0]$, which only involve statistic quantities that can be derived from observational data.

### 2.2 ROBUSTNESS IN CAUSAL INFERENCE

In this work, we aim to enhance the robustness of the ITE model that usually consists of multi-layers neural networks, improving the generalization performance of individual treatment effect estimation. The more rich literature about the robuestness and lipschitz constants of neural networks are presented

in Appendix, and interested readers can refer to it. We consider a $l$-layer feed-forward neural network $f(x) : \mathbb{R}^{n_0} \to \mathbb{R}^{n_{l+1}}$, employed to extract the representations of units in estimating ITE, which was described by the following recursive equations:

$$x^0 = x, \ x^{k+1} = \phi(W^k x^k + b^k) \text{ for } k = 0, ..., l-1, \ f(x) = W^l x^l + b^l \tag{2}$$

where $x \in \mathbb{R}^{n_0}$ is an unit features to the network and $\phi$ denotes the activation functions. $W^k x^k + b^k$, $W^k \in \mathbb{R}^{n_{k+1} \times n_k}$ and $b^k \in \mathbb{R}^{n_{k=1}}$ are the transformation function, weight matrix and bias vector for the $k$-th layer respectively. From the network structure we consider, for each layer transformation and the corresponding takes values, the Lipschitz constant denoted by $\Lambda_k$ with respect to $p$-norm, satisfies:

$$||W^k x^k - W^k \tilde{x}^k||_p \leq \Lambda_k ||x^k - \tilde{x}^k||_p \tag{3}$$

where $x, \tilde{x} \in \mathbb{R}^{n_0}$ are any unit features.

According to the composition rules in estimating the Lipschitz constants Tsuzuku et al. (2018), The Lipschitz constant of $f$ denoted by $\Lambda_p$ satisfies:

$$\Lambda_p \leq \prod_{k=1}^{l} \Lambda_k \tag{4}$$

Thus the Lipschitz constant of the network $f$ can grow exponentially along with its depth $l$. We now give the loss function used in ITE model. Assume there exists a function $L : \mathcal{Y} \times \mathcal{Y} \to \mathbb{R}$ that measures the loss of $f$ on an example $(x, y)$. A common choice for $L$ in causal inference is $|| \cdot ||_p$ loss:

$$L(y, f(x)) = ||y - f(x)||_p \tag{5}$$

For $|| \cdot ||_p$ loss function, the Lipschitz constant denoted by $\lambda_p$ satisfies:

$$||L(y, z) - L(y, \tilde{z})||_p \leq \lambda_p ||z - \tilde{z}||_p, \ \ \forall z, \tilde{z} \in \mathbb{R}, \forall y \in \mathcal{Y} \tag{6}$$

The arguments that we develop above depend only on the Lipschitz constant of the loss, with respect to the norm of interest. And we can directly derive that $\lambda_p \leq 1$. Throughout this paper, we employ the squared loss as our loss function.

## 2.3 ADVERSARIAL EXAMPLES

In practice, given an unit's features $x$ and the corresponding factual outcome $y$, an adversarial example is defined as $\tilde{x} = x + \delta_x$ where $\delta_x$ is a small enough perturbation or error to the original features $x$. As we demonstrated in above example, the contaminated and nearly undistinguishable ECG data would result in predicting incorrect treat effects with high-confidence by the ITE model for heart patients. Therefore, under observational studies, robustness model for ITE estimation is vary important and indispensable. For the parameters and structure $f(\cdot, W)$ of the ITE model, the adversarial example with respect to $p$-norm is formally defined as:

$$\tilde{x} = \underset{||\tilde{x}-x||_p \leq \epsilon}{\arg\max} \ L(f(\tilde{x}, W), y) \tag{7}$$

where $\epsilon$ is to control the perturbation radius. In other words, the strength of the adversary goes down as $\epsilon$ becomes smaller. For an extreme case, when we set $\epsilon$ to 0, the adversarial example $\tilde{x}$ returns to $x$. It is a non-trivial problem to reduce the perturbation to the original exmple in real-world settings. Our goal is to eliminate the impact of adversarial examples to ITE model. By doing that, we provide an informative generalization errors bound with respect to the Lipschitz constant of ITE model, and then propose two types of regularizations called Lipschitz Regularization and reproducing kernel Hilbert space Regularization for mitigating the influence of perturbation while encouraging robustness to ITE model .

## 3 THEORETICAL INSIGHTS

In this section, we will list the common definitions and theoretical results in the ITE estimation. Based on that, we make the relationship between robustness to adversarial examples and the Lipschitz constant of the ITE model, and then give our theoretical results. The complete proofs and details are presented in the Appendix.

**Definition 1.** Let $\Phi : \mathcal{X} \to \mathcal{R}$ be a representation function, $f : \mathcal{R} \times \{0, 1\} \to \mathcal{Y}$ be an hypothesis predicting the outcome of a unit's features $x$ given the representation covariates $\Phi(x)$ and the treatment

assignment $t$. Let $L : \mathcal{Y} \times \mathcal{Y} \to \mathbb{R}_+$ be a loss function. The expected factual and Counterfactual losses of $\Phi$ and $f$ are:

$$\epsilon_F(f, \Phi) = \int_{\mathcal{X} \times \mathcal{T} \times \mathcal{Y}} L(y, f(\Phi(x), t)) p(x, t, y) dx dt dy$$

$$\epsilon_{CF}(f, \Phi) = \int_{\mathcal{X} \times \mathcal{T} \times \mathcal{Y}} L(y, f(\Phi(x), 1 - t)) p(x, 1 - t, y) dx dt dy$$
(8)

It can be seen that $\epsilon_F$ measures how well do $f$ and $\Phi$ predict the factual outcomes based on unit features and treatment sampled from the same distribution as our data sample. While $\epsilon_{CF}$ aims to measure the counterfactual outcomes based on the same unit features but the opposite treatment.

**Definition 2.** The expected factual treated and control losses are:

$$\epsilon_F^{t=1}(f, \Phi) = \int_{\mathcal{X} \times \mathcal{Y}} L(y, f(\Phi(x), 1)) p(x, y | T = 1) dx dy$$

$$\epsilon_F^{t=0}(f, \Phi) = \int_{\mathcal{X} \times \mathcal{Y}} L(y, f(\Phi(x), 0)) p(x, y | T = 0) dx dy$$
(9)

Accordingly, we can obtain an immediate results $\epsilon_F(f, \Phi) = p(t = 1)\epsilon_F^{t=1}(f, \Phi) + p(t = 0)\epsilon_F^{t=0}(f, \Phi)$.

**Definition 3.** For the adversarial examples, the expected factual loss of $f$ and $\Phi$ is :

$$\epsilon_{Fadv}(f, \Phi, \epsilon) = \int_{\mathcal{X} \times \mathcal{T} \times \mathcal{Y}} \max_{||\tilde{x} - x||_p \leq \epsilon} L(y, f(\Phi(\tilde{x}), t)) p(x, t, y) dx dt dy$$
(10)

It can be seen that $\epsilon_{Fadv}$ measures how well our prediction with $f$ and $\Phi$ would do if the inputs are replaced by the adversarial examples. According to these definitions, we now give the generalization bound of the expected factual loss of $f$ and $\Phi$ with respect to the adversarial examples,

**Theorem 1.** *Let $\epsilon$ denotes the strength of the adversary. Let $\lambda_p$ denotes the Lipschitz constant of $L$ loss and $\Lambda_p$ stands for the Lipschitz constant of $f$ and $\Phi$ defined in Definition 1, then we have:*

$$\epsilon_{Fadv}(f, \Phi, \epsilon) \leq \epsilon_F(f, \Phi) + \lambda_p \Lambda_p \epsilon$$

*Remark.* Theorem 1 provides an upper bound for the $L$ loss with the Lipschitz constant based on adversarial samples, which suggests that the vulnerability of ITE model to adversarial examples can be controlled by its Lipschitz constant.

**Definition 4.** The estimation of treatment effect by an hypothesis $f$ and a representation function $\Phi$ for unit $x$ is:

$$\hat{\tau}(x) = f(\Phi(x), 1) - f(\Phi(x), 0)$$
(11)

**Definition 5.** The expected Precision in Estimation of Heterogeneous Effect (PEHE) Hill (2011) loss of $f$ and $\Phi$ is:

$$\epsilon_{PEHE}(f) = \int_{\mathcal{X}} (\hat{\tau}(x) - \tau(x))^2 p(x) dx$$
(12)

**Definition 6.** Integral Probability Metric (IPM). For two probability density functions $p, q$ defined over $\mathcal{S} \in \mathbb{R}^d$, and for a function family $G$ of functions $g : \mathcal{S} \to \mathbb{R}$, The IPM is Shalit et al. (2017):

$$\text{IPM}_G(p, q) := \sup_{g \in G} \left| \int_{\mathcal{S}} g(s)(p(s) - q(s)) ds \right|$$
(13)

From the definition we can see that IPM measures the distance between two distributions. For rich enough function families $G$, IPM is a true metric over the corresponding set of probabilities Shalit et al. (2017); Qin et al. (2021). When we let $G$ satisfy the family of 1-Lipschitz functions, i.e., $G = \{g : ||g||_p \leq 1\}$ we obtain the Wasserstein distance denote by $Wass_G(\cdot, \cdot)$ between distributions. While $G = \{g \in \mathcal{H} \ s.t. \ ||g||_{\mathcal{H}} \leq 1\}$, we derive Maximum mean discrepancy denote by $MMD_G(\cdot, \cdot)$ between distributions. Where $\mathcal{H}$ represents a reproducing kernel Hilbert space(RKHS) Sriperumbudur et al. (2009). In the rest of the paper, we consider an estimation for ITE in the form of $f(\Phi(x), 1) - f(\Phi(x), 0)$. Next is the generalization bound for PEHE loss derived in Shalit et al. (2017).

**Proposition 1.** *Shalit et al. (2017). Let $\Phi : \mathcal{X} \to \mathcal{R}$ be a one-to-one representation function and $f : \mathcal{R} \times \mathcal{T} \to \mathcal{Y}$ be an hypothesis. Let $G$ be a family of functions $g : \mathcal{R} \to \mathcal{Y}$. Assume that there*

*exists a $\ell_2$ loss, $L : \mathcal{Y} \times \mathcal{Y} \to \mathcal{R}_+$, and a constant $C_\Phi > 0$, such that for fixed $t \in \{0, 1\}$, the per-unit expected loss function $\ell_{f,\Phi}(x, t) = \int_{\mathcal{Y}} L(Y_t, f(\Phi(x), t)) p(Y_t|x) dY_t$ obey $\frac{1}{C_\Phi} \cdot \ell_{f,\Phi}(x, t) \in G$. Then,*

$$\epsilon_{PEHE}(f, \Phi) \leq 2(\epsilon_{CF}(f, \Phi) + \epsilon_F(f, \Phi) - C_Y)$$
$$\leq 2\left(\epsilon_F^{t=0}(f, \Phi) + \epsilon_F^{t=1}(f, \Phi)\right) + 2\left(C_\Phi \cdot IPM_G(p_\Phi^{t=1}, p_\Phi^{t=0}) - C_Y\right)$$

*where $p_\Phi^{t=1} = p(\Phi(x)|t = 1)$, $p_\Phi^{t=0} = p(\Phi(x)|t = 0)$ are the treated and control distributions define over $\mathcal{R}$ separately, and $C_Y$ is a constant induced over the variance of the outcomes $Y_t$.*

*Remark.* Proposition 1 provides an upper bound for the expected ITE estimation error of a representation, which is bounded by the sum of the standard regression generalization error on treated and control groups and the distance between the treated and control distributions. In practice, the distance is measured by MMD. While the promising explicit bounds for MMD or WASS distances are derived by the authors, the rules for the function family $G$ are not satisfied to transform IPM to MMD or WASS distances metrics without considering the Lipschitz constant of $f$ and $\Phi$ in estimation of ITE. To alleviate this problem, we first derive a generalization bound for the expected factual loss, and further derive the new generalization bound for the expected ITE estimation error. In order to use IPM to measure the distances between treated and control distributions, we also obtain two explicit bounds for WASS and MMD distances respectively. We find that all the bounds are bounded by one of the most important terms, the Lipschitz constant of ITE model. The results are presented as follows.

**Theorem 2.** *Let $C_p(\mathcal{D}, \gamma)$ be the covering number of $\mathcal{D}$ using $\gamma$-balls for $|| \cdot ||_p$. Let $C_d = \sup_{x,t,W,y} L(y, f(\Phi(x), t))$, where $W$ is the parameters of $f$ and $\Phi$. Then for any $\delta > 0$, with probability at least $1 - \delta$ over the i.i.d. samples $\{(x_i, t_i, y_i)\}_{i=1}^m$, we have:*

$$\epsilon_F(f, \Phi) \leq \frac{1}{m} \sum_{i=1}^m L(y_i, f(\Phi(x_i), t_i)) + \lambda_p \Lambda_p \gamma + C_d \sqrt{\frac{2C_p(\mathcal{D}, \gamma) \ln 2 + 2 \ln(1/\delta)}{m}}$$

*Remark.* Theorem 2 provide an upper bound for the expected factual loss that is crucial for bounding the ITE estimation error demonstrated in Proposition 1. This bound indicates that the Lipschitz constant of $f$ and $\Phi$ can control the difference between the average empirical factual loss on the training set and the generalization performance. In addition, the covering numbers of $\lambda$-balls tends to increase exponentially with the dimension of unit features $x$ becomes larger. Thus, the bound above show that it is critical to reduce the Lipschitz constant of ITE model for both good generalization and robustness to adversarial examples. In the following, we now give the informative generalization bound for the expected ITE estimation error.

**Theorem 3.** *Under the conditions of Definition 1, Proposition 1 and Theorem 2, with probability at least $1 - \delta$,*

$$\epsilon_{PEHE}(f, \Phi) \leq \frac{4}{m} \sum_{i=1}^m L(y_i, f(\Phi(x_i), t_i))$$
$$+ 4\left(\lambda_p \Lambda_p \lambda + C_d \sqrt{\frac{2C_p(\mathcal{D}, \gamma) \ln 2 + 2 \ln(1/\delta)}{m}}\right) + 2\left(C_\Phi \cdot IPM_G(p_\Phi^{t=1}, p_\Phi^{t=0}) - C_Y\right)$$

*Remark.* Theorem 3 provides an upper bound for PEHE loss , which mainly consists of the empirical regression losses on training set, the Lipschitz constant of $f$ and $\Phi$, the covering numbers of $\lambda$-balls and the distance between the treated and control distributions induced by $\Phi$. Except for the Lipschitz constant term, all of which can be empirically estimated or approximated by some deep learning tricks. The upper bound decreases as the Lipschitz constant $\Lambda_p$ gets small. Therefore, Theorem 3 instructs us to constrain the parameters of $f$ and $\Phi$ that can obtain a robust ITE model with a small Lipschitz constant, even less than or equal to 1.

**Theorem 4.** *Let $\Phi : \mathcal{X} \to \mathcal{R}$ be a one-to-one representation function and $f : \mathcal{R} \times \mathcal{T} \to \mathcal{Y}$ be an hypothesis. Let $G$ be a family of functions $g : \mathcal{R} \to \mathcal{Y}$. Assume that there exists a $\ell_2$ loss, $L : \mathcal{Y} \times \mathcal{Y} \to \mathcal{R}_+$, $\ell_{f,\Phi}(x, t) \in G$ for $t = 0, 1$. if the Lipschitz constant of $\ell_{f,\Phi}(x, t) \in G$ is upper*

*bounded by* 1, *then,*

$$\epsilon_{PEHE}(f, \Phi) \leq \frac{4}{m} \sum_{i=1}^{m} L\left(y_i, f(\Phi(x_i), t_i)\right)$$

$$+ 4\left(\lambda_p \Lambda_p \lambda + C_d \sqrt{\frac{2C_p(\mathcal{D}, \gamma)\ln 2 + 2\ln(1/\delta)}{m}}\right) + 2\left(WASS_G(p_\Phi^{t=1}, p_\Phi^{t=0}) - C_Y\right)$$

*if the function space $G$ satisfies $G = \{g \in \mathcal{H} \ s.t. \ ||g||_{\mathcal{H}} \leq 1\}$, in which $\mathcal{H}$ be a reproducing kernel Hilbert space, then,*

$$\epsilon_{PEHE}(f, \Phi) \leq \frac{4}{m} \sum_{i=1}^{m} L\left(y_i, f(\Phi(x_i), t_i)\right)$$

$$+ 4\left(\lambda_p \Lambda_p \lambda + C_d \sqrt{\frac{2C_p(\mathcal{D}, \gamma)\ln 2 + 2\ln(1/\delta)}{m}}\right) + 2\left(MMD_G(p_\Phi^{t=1}, p_\Phi^{t=0}) - C_Y\right)$$

*Remark.* In Theorem 4, the upper bound for PEHE loss is similar to the bound derived in Theorem 3, except for the distance metric. Actually, we use WASS and MMD distance metric to substitute for the IPM distance metric, repsectively, while deriving a lower bound. In order to employ the specific distance metrics to balance the distribution between treated and control groups and lowering the bounds of PEHE loss, we propose two types of regularizations called Lipschitz Regularization and RKHS Regularization. The former is able to provide a appropriate functions family for WASS distance, i.e., $G = \{g : ||g||_p \leq 1\}$, while reducing the Lipschitz constant of $f$ and $\Phi$, and the latter can give the a satisfactory functions family for MMD distance, i.e., $G = \{g \in \mathcal{H} \ s.t. ||g||_{\mathcal{H}} \leq 1\}$.

## 4 THE PROPOSED METHOD

In this section, we first give the details about the two types of regularizations and then introduce the algorithm for ITE estimation.

### 4.1 LIPSCHITZ REGULARIZATION

According to Theorem 4, in order to guarantee the WASS distance works well while controlling Lipschitz constant of ITE model, we need to maintain the spectral norm of the weight matrix of each transformation layer at 1. We now give the Lipschitz constants of standard layers as a function of their parameters.

**Definition 7.** The Lipschitz constant for each transformation layer is:

$$||W^k||_p = \sup_{||z||_p=1} ||W^k z||_p \tag{14}$$

where $||W^k||_2$ is the maximum singular value $W^k$, which called the spectral norm of $W^k$ . Then for the $l$-layer feed-forward neural network, its Lipschitz constant satisfies:

$$\Lambda_p \leq \prod_{k=1}^{l} ||W^k||_p \tag{15}$$

Motivated by the works of parseval tightness theory in Kovačević et al. (2008); Cisse et al. (2017),which demonstrates that the orthonormality of weight matrices are sufficient to control the spectral norm. we aim to constrain the parameters with orthonormality for each transformation layer:

$$\Re_k(f) = \frac{\beta}{2}||W^{k^T}W^k - I||_2^2 \tag{16}$$

where $I$ refers to the identity matrix. The gradient of this regularization term is $\nabla_{W^k}\Re_k(f) = \beta(W^k W^{k^T} - I)W^k$. Consequently, we perform this gradient in each update step to update the parameters $W^k$. For the $l$-layer feed-forward neural network $f$, we constrain:

$$\Re(f) = \sum_{k=1}^{l} \Re_k(f) \tag{17}$$

which is regarded as a Lischits regularization to optimization objective in ITE estimation problem. With the orthogonality constrains, the Lipschitz constant of each transformation layer is then approximately less than or equal to 1.

## 4.2 RKHS Regularization

According to Theorem 4, we aim to guarantee the MMD distance metric works well in balancing the distributions by a representation between the treated and control groups. Consequently, we perform a product constrains to the outputs of the last transformation layer. Formally, product constrains is defined as:

$$\Re(f) = \frac{\beta}{2}||\ell_{f,\Phi}(x,t)||_{\mathcal{H}} - 1 \tag{18}$$

where $\mathcal{H}$ reprensents the RKHS. The gradient of this regularization term is $\beta\ell_{f,\Phi}(x,t)$. In practice, we update the parameters of $f$ and $\Phi$ in a chain rules by the last outputs of $f$. With the RKHS regularization, the functions family $G$ are then approximately satisfied for MMD distance metric.

## 4.3 Algorithm for Estimating ITE

According to the above theoretical analysis in Section 3, we propose a framework called RITE to minimize the upper bounds in Theorem 3 and Theorem 4. We follow the commonly used form of objective function to measure the ITE. The optimization problem in our framework is shown as the following:

$$\min_{f,\Phi} \frac{1}{m}\sum_{i=1}^{m} w_i \cdot L(y_i, f(\Phi(x_i),t_i)) + \beta \cdot \Re(f) + \alpha \cdot IPM_G(\hat{p}_{\Phi}^{t=1}, \hat{p}_{\Phi}^{t=0})$$
$$s.t \quad w_i = \frac{t_i}{2u} + \frac{1-t_i}{2(1-u)}, \quad where \quad u = \frac{1}{m}\sum_{i=1}^{m} t_i \tag{19}$$

where $u = p(t = 1)$ is simply the proportion of treated units in the population, the weights $w_i$ compensates for the difference in treatment group size Shalit et al. (2017), $\hat{p}_{\Phi}^{t=1}$ and $\hat{p}_{\Phi}^{t=0}$ are learned high-dimensional representation for treated and control groups respectively. Note that $IPM_G(\cdot,\cdot)$ is the distance metric and the specific implementations of it depends on the proposed regularization $\Re(f)$. If we let $\Re(f)$ Lipschitz regularization, then $IPM_G(\cdot,\cdot)$ becomes WASS distance metric. If we set $\Re(f)$ to RKHS regularization, we obtain the MMD distance metric. We refer to the model minimizing equation 19 with Lipschitz regularization as $RITE_{WASS}$ and the variant with RKHS regularization as $RITE_{MMD}$. Both models are trained by the adaptive moment estimation (Adam) Kingma & Ba (2014). The details are described in Appendix.

## 5 Experiments

### 5.1 Experiment Setup

ITE estimation is more difficult compared to machine learning tasks, the reason is that we rarely have access to ground-truth treatment effect in real-world scenario. In order to measure the proposed framework, we conduct experiments based on one synthetic examples, **Sim**, and two standard benchmark datasets, **ACIC** Dorie et al. (2019) and **IHDP** Dorie et al. (2019). We compare our model with the following 11 representative baselines: Random Forests (RF) Breiman (2001), Causal Forests (CF) Wager & Athey (2018), Causal Effect Variational Autoencoder (CEVAE) Louizos et al. (2017), DragonNet Shi et al. (2019), Meta-Learner algorithms S-Learner Nie & Wager (2021) and T-Learner Künzel et al. (2019), Balancing Neural Network (BNN) Johansson et al. (2016), Treatment-Agnostic Representation Network (TARNet) Shalit et al. (2017) as well as Counterfactual Regression with the Wasserstein metric ($CFR_{WASS}$) Shalit et al. (2017) and the squared linear MMD metric ($CFR_{MMD}$) Shalit et al. (2017), along with a extension of CRF method Query-based Heterogeneous Treatment Effect estimation (QHTE) Qin et al. (2021). The detailed description about implementations and datasets are shown in Appendix. One also can find more details about the implementation of all adopted baselines and our methods and full experimental settings at https://github-rite.github.io/rite/.

Table 1: Individual treatment effect estimation on ACIC, IHDP and Sim test set. The top module consists of baselines from recent works. The bottom module consists of our proposed methods. In each module, we present each of the result with form mean ± standard deviation and we use bold fonts to label the best performance. Lower is better.

| Datasets | ACIC | | IHDP | | Sim | |
|---|---|---|---|---|---|---|
| Metric | $\sqrt{\epsilon_{PEHE}}$ | $\epsilon_{ATE}$ | $\sqrt{\epsilon_{PEHE}}$ | $\epsilon_{ATE}$ | $\sqrt{\epsilon_{PEHE}}$ | $\epsilon_{ATE}$ |
| RF | 3.09 ± 1.48 | 1.16 ± 1.40 | 4.61 ± 6.56 | 0.70 ± 1.50 | 3.36 ± 0.01 | 3.87 ± 0.01 |
| CF | 1.86 ± 0.73 | 0.28 ± 0.27 | 4.46 ± 6.53 | 0.81 ± 1.36 | 1.81 ± 0.04 | 0.08 ± 0.06 |
| S-learner | 3.86 ± 1.45 | 0.41 ± 0.35 | 5.76 ± 8.11 | 0.96 ± 1.8 | 1.92 ± 0.05 | 0.06 ± 0.05 |
| T-learner | 2.33 ± 0.86 | 0.79 ± 0.68 | 4.38 ± 7.85 | 2.16 ± 6.17 | 0.57 ± 0.02 | **0.03 ± 0.02** |
| CEVAE | 5.63 ± 1.58 | 3.96 ± 1.37 | 7.87 ± 7.41 | 4.39 ± 1.63 | 1.92 ± 0.05 | 0.12 ± 0.18 |
| BNN | 2.00 ± 0.86 | 0.43 ± 0.36 | 3.17 ± 3.72 | 1.14 ± 1.7 | 1.08 ± 0.09 | 0.26 ± 0.15 |
| DragonNet | 1.26 ± 0.32 | 0.15 ± 0.13 | 1.46 ± 1.52 | 0.28 ± 0.35 | 0.43 ± 0.05 | 0.09 ± 0.07 |
| TARNet | 1.30 ± 0.46 | 0.15 ± 0.12 | 1.49 ± 1.56 | 0.29 ± 0.40 | 0.45 ± 0.04 | 0.09 ± 0.06 |
| CFR$_{MMD}$ | 1.24 ± 0.31 | 0.17 ± 0.14 | 1.51 ± 1.66 | 0.3 ± 0.52 | 0.46 ± 0.04 | 0.09 ± 0.07 |
| CFR$_{WASS}$ | 1.27 ± 0.38 | 0.15 ± 0.12 | 1.43 ± 1.61 | 0.27 ± 0.41 | 0.49 ± 0.05 | 0.10± 0.07 |
| QHTE | 1.32 ± 0.41 | 0.19 ± 0.18 | 1.83 ± 1.9 | 0.34 ± 0.43 | 0.51 ± 0.06 | 0.18 ± 0.06 |
| RITE$_{MMD}$ | 1.00 ± 0.23 | 0.15 ± 0.12 | **0.45 ± 0.06** | **0.12 ± 0.09** | 0.44 ± 0.06 | 0.06 ± 0.04 |
| RITE$_{WASS}$ | **0.97 ± 0.28** | **0.14 ± 0.11** | 1.34 ± 1.53 | 0.28 ± 0.39 | **0.37 ± 0.06** | 0.06 ± 0.05 |

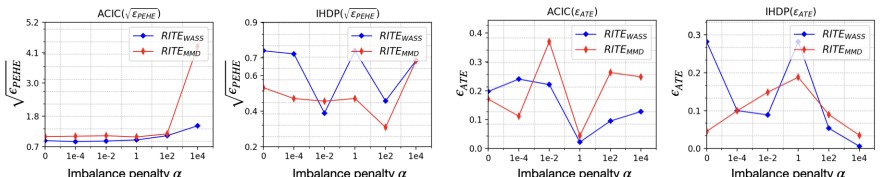

Figure 1: Influence of the imbalance penalty $\alpha$ on our model performance in terms of $\sqrt{\epsilon_{PEHE}}$ and $\epsilon_{ATE}$. The performances of different distance metric implementations are labeled with different colors. Lower is better.

## 5.2 OVERALL RESULTS

The overall comparison results are presented in Table 1, from which we can see: Compared to the synthetic datasets, the performance of all the models are a little higher on real-world benchmark datasets, which is because of the imbalanced distribution nature between treated and control groups , and verifies the difficulties of the ITE estimation task itself. Representation learning methods like DragonNet can usually obtain better performance than the traditional machine learning method like RF, which agrees with the previous work Qin et al. (2021); Shalit et al. (2017), and verifies the usefulness of predicting the ITE by a deep neural network. Among representation learning models, the best performance is usually achieved when the model is based on the IPM distance metric. This is as expected, since the IPM distance metric based on the studied representation can effectively reduce the distribution shift between treated and control groups, improving the generalization performance of ITE estimation. Encouragingly, our model can achieve the best performance on all the metrics across different datasets, where the improvements are mostly significant. The result is consistent to our theoretical analysis in section 3. Comparing with the baselines, we introduce the Lipschitz regularization and RKHS regularization separately to reduce the Lipschitz constant of ITE model, improving the generalization performance of ITE estimation. Between the different implementations of IPM distance metric, we find that WASS is a little superior than MMD in most cases. We speculate that WASS is more suitable for balancing the representation distributions, which can be more appropriate for the real-world datasets.

## 5.3 ROBUSTNESS CERTIFICATION

In this section, we aim to demonstrate the validity of the robustness of our model from the perspective of experiment. In order to achieve a fair comparison performance, we adopt the representative deep learning methods BNN, DragonNet, TARNet, CFR$_{WASS}$, CFR$_{MMD}$, and our methods to conduct this experiments. More concretely, for given a test data point $x$, we generate a new one $x' = x + \delta_X$ to substitute for the original test data point. In practice, we add noise in $\{\mathbb{U}(-1, 1)\}^{dim(x)}$ to each data point. The results are presented in Table 2. By imposing a small perturbation values on the input point, we can find that all of the performance across dataset ACIC and IHDP have been degraded

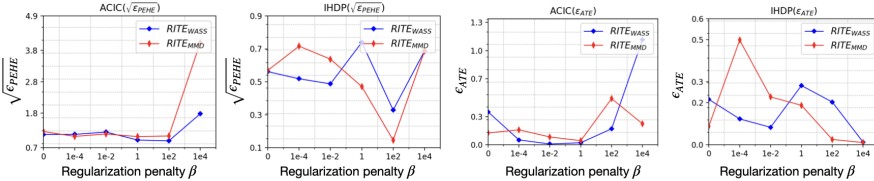

Figure 2: Influence of the regularization penalty $\beta$ on our model performance in terms of $\sqrt{\epsilon_{PEHE}}$ and $\epsilon_{ATE}$. The performances of different distance metric implementations are labeled with different colors. Lower is better.

Table 2: Performance comparison between the model testing in the original test sets and with small perturbation in test sets. We use "X-Noisy" to represent the test set with noisy when the model is "X". We highlight the best performance with bold fonts. Lower is better.

| Datasets | ACIC | | IHDP | |
|---|---|---|---|---|
| Metric | $\sqrt{\epsilon_{PEHE}}$ | $\epsilon_{ATe}$ | $\sqrt{\epsilon_{PEHE}}$ | $\epsilon_{ATe}$ |
| BNN-Noisy | $4.84 \pm 0.15$ | $1.50 \pm 0.46$ | $3.25 \pm 0.39$ | $3.03 \pm 0.55$ |
| DragonNet-Noisy | $1.55 \pm 0.10$ | $0.21 \pm 0.16$ | $0.65 \pm 0.08$ | $0.15 \pm 0.11$ |
| TARNet-Noisy | $1.48 \pm 0.10$ | $0.22 \pm 0.17$ | $0.67 \pm 0.08$ | $0.17 \pm 0.11$ |
| CFR$_{MMD}$-Noisy | $1.48 \pm 0.10$ | $0.21 \pm 0.16$ | $0.67 \pm 0.08$ | $0.16 \pm 0.12$ |
| CFR$_{WASS}$-Noisy | $1.52 \pm 0.10$ | $0.23 \pm 0.16$ | $0.63 \pm 0.08$ | $0.17 \pm 0.12$ |
| RITE$_{MMD}$-Noisy | $1.17 \pm 0.07$ | $0.16 \pm 0.12$ | $\mathbf{0.48 \pm 0.06}$ | $\mathbf{0.12 \pm 0.09}$ |
| RITE$_{WASS}$-Noisy | $\mathbf{1.04 \pm 0.06}$ | $\mathbf{0.15 \pm 0.11}$ | $0.56 \pm 0.08$ | $0.16 \pm 0.11$ |

jointly comparing with Table 1. It is encouraging to see that our framework can still outperform the base models in all cases. This observation suggests that our framework can indeed improve the model robustness even if the input points have been perturbed. For our framework, the strategies of Lipschitz regularization and RKHS regularization seem to have different advantages under different settings, and they alternatively achieve the best performances, which is analogous to the results observed in Table 1. Based on this observation, we speculate that, for larger datasets application scenarios, the CFR$_{WASS}$ method can be leveraged to build more robust treatment effect model. Otherwise, the CFR$_{MMD}$ may also be competitive.

## 5.4 PARAMETER STUDY

As detailed in the above sections, our main optimization objective is composed of many terms. Readers may be interest in how different terms contribute the final performance. In order to answer this question and illustrate the influence of different terms, in this section, we conduct the parameter studies, where the hyper-parameters settings follow the above experiments and we compare our model by varying the imbalance penalty $\alpha$ and regularization penalty $\beta$. For optimization objective ( 19), the regularization influence will decrease when the regularization penalty $\beta$ becomes smaller. It is analogous to the distance metric IPM term. We tune $\alpha$ and $\beta$ both in [0,1e-4,1e-2,1,1e2,1e4]. The results are presented in Figure 1 and 2. We can see: the best performance is usually achieved when $\alpha$ and $\beta$ is moderate. This agrees with our opinion in section 3, i.e., too small $\alpha$ and $\beta$ may introduce too imbalance representation into the training process, while too large $\alpha$ and $\beta$ may severely impact the predictions made by the ITE model . By tunning $\alpha$ and $\beta$ in proper ranges, we are allowed to achieve better trade-offs to improve the ITE estimation performance.

## 6 CONCLUSION

In this paper, we propose to enhance robustness and generalization performance in estimating ITE by adversarial machine learning. To achieve this goal, we first theoretically analyze the bound of the PEHE loss, and then design two types of regularizations for encouraging robustness in estimating ITE according to specific distance metric. In the experiments, we evaluate our framework based on both synthetic and semi-synthetic datasets to demonstrate its effectiveness and generality. This paper makes a first step on applying the idea of adversarial machine learning to the field of estimating ITE. There is still much room for improvement. To begin with, one can incorporate more sophisticated model to extract the representation of covariates, and at the same time devise effective mechanism for encouraging robustness to causal inference. In addition, in order to reduce the time-comsuming, people can also choose more specific layers to constrain its parameters.

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

# A APPENDIX

## A.1 PROOF OF THEORY 1

*Proof.* For the expected factual loss of $\Phi$ and $f$ over the adversarial samples , we have:

$$\epsilon_{fadv} - \epsilon_F \leq |\epsilon_{fadv} - \epsilon_F|$$

and hence,

$$
\begin{aligned}
\epsilon_{fadv} &\leq \epsilon_F + |\epsilon_{fadv} - \epsilon_F| \\
&\leq \epsilon_F + \int_{\mathcal{X} \times \mathcal{T} \times \mathcal{Y}} \max_{||\tilde{x}-x||_p \leq \epsilon} |L(y, f(\Phi(\tilde{x}), t)) \\
&\quad - L(y, f(\Phi(x), t))| \, p(x, t, y) dx dt dy \\
&\leq \epsilon_F + \lambda_p \Lambda_p \epsilon
\end{aligned}
$$

$\square$

## A.2 PROOF OF THEORY 2

*Proof.* We reformulate the expected factual loss of $\Phi$ and $f$ as:

$$\epsilon_F(f, \Phi) = \mathbb{E}_{(x,t,y) \sim \mathcal{D}}[L(y, f(\Phi(x)), t)]$$

and its empirical factual loss is:

$$\hat{\epsilon}_F(f, \Phi) = \frac{1}{m} \sum_{i=1}^{m} L(y_i, f(\Phi(x_i), t_i))$$

Let $C_p(\mathcal{D}, \gamma)$ be the covering number of $\mathcal{D}$ using $\gamma$-balls for $|| \cdot ||_p$. In our paper, we focus on the binary treatment case where $t \in \{0, 1\}$. Therefore, we can partition $\mathcal{D}$ into $2\mathcal{N}(\gamma/2, \mathcal{X}, || \cdot ||_p) \times \mathcal{N}(\gamma/2, \mathcal{Y}, || \cdot ||_p)$ subsets where $\mathcal{N}(\gamma/2, \mathcal{X}, || \cdot ||_p)$ is the $\gamma/2$-covering number of $\mathcal{X}$ and $\mathcal{N}(\gamma/2, \mathcal{Y}, || \cdot ||_p)$ is the $\gamma/2$-covering numer of $\mathcal{Y}$. For two samples $x_1$ and $x_2$ who belong to a same subset $\mathcal{D}_i$, then we have $||x_1 - x_2||_p \leq \gamma$, and the corresponding outcomes $y_1$ and $y_2$ satifies: $||y_1 - y_2||_p \leq \gamma$.

**Definition 8.** Let $K$ be the covering numer of $\mathcal{D}$ using $\gamma$-balls for $|| \cdot ||_p$ and $\{\mathcal{D}_1, ..., \mathcal{D}_K\}$ be the partitioned subsets of $\mathcal{D}$ as defined above. Let $D = \{(x_i, t_i, y_i)\}_{i=1}^{m}$ be the observational data. Let $N_i$ be the set of index of points of the observational sample $(x, t, y)$ that fall into the $\mathcal{D}_i$. Note that $\{|N_1|, ..., |N_K|\}$ is an IID multinomial random variable with parameters $m$ and $\{\mu(\mathcal{D}_1), ..., \mu(\mathcal{D}_K)\}$. By the Breteganolle-Huber-Carol inequality Xu & Mannor (2012), the following holds with probability at least $1 - \delta$:

$$\sum_{i=1}^{K} \left| \frac{|N_i|}{m} - \mu(\mathcal{D}_i) \right| \leq \sqrt{\frac{2K \ln 2 + 2\ln(1/\delta)}{m}}$$

Then, We have

$$|\epsilon_F(f, \Phi) - \hat{\epsilon}_F(f, \Phi)|$$

$$= \left| \sum_{i=1}^{K} \mathbb{E}\left[L(y, f(\Phi(x), t)) | (x, t, y) \in \mathcal{D}_i\right] \mu(\mathcal{D}_i) - \frac{1}{m} \sum_{i=1}^{m} L(y_i, f(\Phi(x), t_i)) \right|$$

$$\leq \left| \sum_{i=1}^{K} \mathbb{E}\left[L(y, f(\Phi(x), t)) | (x, t, y) \in \mathcal{D}_i\right] \frac{|N_i|}{m} - \frac{1}{m} \sum_{i=1}^{m} L(y_i, f(\Phi(x), t_i)) \right|$$

$$+ \left| \sum_{i=1}^{K} \mathbb{E}\left[L(y, f(\Phi(x), t)) | (x, t, y) \in \mathcal{D}_i\right] \mu(\mathcal{D}_i) \right.$$

$$\left. - \sum_{i=1}^{K} \mathbb{E}\left[L(y, f(\Phi(x), t)) | (x, t, y) \in \mathcal{D}_i\right] \frac{|N_i|}{m} \right|$$

$$\leq \left| \frac{1}{m} \sum_{i=1}^{K} \sum_{j \in N_i} \max_{(x, t, y) \in \mathcal{D}_i} |L(y_j, f(\Phi(x_j), t_j)) - L(y, f(\Phi(x), t))| \right|$$

$$+ \left| \max_{(x, t, y) \in \mathcal{D}} |L(y, f(\Phi(x), t))| \sum_{i=1}^{K} \left| \frac{|N_i|}{m} - \mu(\mathcal{D}_i) \right| \right|$$

$$\leq \lambda_p \Lambda_p \epsilon + \mathcal{C}_d \sum_{i=1}^{K} \left| \frac{|N_i|}{m} - \mu(\mathcal{D}_i) \right|$$

By integrating Definition 8, the proof of Theorem 2 is done.

$\square$

### A.3 PROOF OF THEORY 3

*Proof.* The result is derived by bounding the two $\epsilon_F^t(f, \Phi)$ terms in Proposition 1 with the inequality in Theorem 2. $\square$

### A.4 RELATED WORK

**Estimating individual treatment effect.** How to effectively and correctly measure individual treatment effect has recently attracted increasing attention from the research community. It basically aims to discover the underlying patterns of the distribution between treated and control group. To model this character, early methods are based on re-weighting methods Austin (2011); Imai & Ratkovic (2014); Fong et al. (2018) that is an effective approach to overcome the selection bias induced by the existence of covariates in observational studies. Another widely used techniques for individual treatment effect inference are traditional machine learning, including Bayesian Additive Regression Trees (BART) Hill (2011), Random Forests (RF) Breiman (2001), Causal Forests (CF) Wager & Athey (2018), etc. These methods have more flexibility and predictive ability in balancing the distribution between treated and control groups compared to re-weighting methods. In addition, some promising works like S-Learner Nie & Wager (2021) and R-Learner Künzel et al. (2019) are based on meta-learning to utilize any supervised learning or statistical regression methods to estimate ITE. Recent years have witnessed many studies on adapting more sophisticated mechanisms to causal effect inference, and in particular to measure individual level treatment effect. For example, Causal Effect Variational Autoencoder (CEVAE) Louizos et al. (2017) leverage Variational Autoencoders to obtain the unobserved confounders and simultaneously infer causal effects, DragonNet Shi et al. (2019) design three-head components to predict the treatment effects as well as adjust the distribution by a process of inferring treatments. Besides, more cutting-edge mechanism like Integral Probability Metric (IPM) Qin et al. (2021); Johansson et al. (2016) are applied to minimize generalization bound for ITE estimation, which is composed of factual loss and the discrepancy between the treated and control distributions. The representative CFR Shalit et al. (2017) method enforce the similarity between the distributions of treated and control groups in the representation space by a penalty term IPM. While the boundary of estimation of individual treatment effect from observational data has

been pushed by these models, an important problem is still under-explored, that is the robustness of the treatment effect predicted by deep neural networks when their input is subject to an adversarial perturbation. In this paper, we bridge this gap by proposing two types of regularizations called Lipschitz regularization and RKHS regularization to the original causal models for encouraging smoothness as well as improving the generalization performance.

**Adversarial machine learning.** Adversarial machine learning is a concept describing the study of robust machine learning techniques against an adversarial perturbations Huang et al. (2011). In the past few years, in order to facilitate the security and robustness of a model, adversarial machine learning has been widely applied to the machine learning community. For example, Cisse et al. (2017); Virmaux & Scaman (2018); Zhang et al. (2021) incorporated some adversarial examples or robustness regularization into original objective for tackling sensitive issues in neural networks. In addition to that, some works Deldjoo et al. (2021); Tian & Xu (2021) attempt to enhance the robustness of recommender system and audio-visual learning model respectively and simultaneously improve its generalization performance via a way of adversarial optimization framework. Another important application is in computer vision Santurkar et al. (2019); Elsayed et al. (2018), in which the adversarial examples are used to enhance the parameters of original model. We realize the idea of adversarial machine learning in the field of causal inference for estimating individual treatment effect . More importantly, we provide theoretical analysis on the expected precision in estimation of heterogeneous effect (PEHE) loss and design two types of regularizations for encouraging robustness.

## A.5 IMPLEMENTATION DETAILS.

We implement our methods based on QHTE Qin et al. (2021). We use the same set of hyperparameters for RITE across three datasets. More specifically, we adopt 3 fully-connected exponential-linear layers for the representation function $\Phi$ and 3 similar architecture layers for the ITE prediction function $f$. The difference is that layer sizes are 200 for former, and 100 for latter. Batch normalization Ioffe & Szegedy (2015) is applied to facilitate training, and all but the output layer use ReLU (Rectified Linear Unit) Agarap (2018) as activation functions whose Lipschitz constant is less than or equal to 1. In the main optimization objective, we set $\alpha$ and $\beta$ both to 1. The more details about the implementation of all adopted baselines and our methods and full experimental settings are presented at https://github-rite.github.io/rite/. As introduced in section 4, we use the Wasserstein (RITE$_{WASS}$) and the squared linear MMD (RITE$_{MMD}$) distances to penalize imbalance. To overcome the lack of robustness in network for RITE$_{MMD}$ method, we also add the robustness regularization to the main optimization objective. The commonly used metrics including Rooted Precision in Estimation of Heterogeneous Effect (PEHE) Hill (2011) and Mean Absolute Error (ATE) Shalit et al. (2017) are applied for evaluating the quality of individual treatment effects. Formally, they are defined as:

$$\sqrt{\epsilon_{PEHE}} = \sqrt{\frac{1}{n}\sum_{i=1}^{n}(\hat{\tau}_i - \tau_i)^2}, \quad \epsilon_{ATE} = |\frac{1}{n}\sum_{i=1}^{n}(\hat{\tau}) - \frac{1}{n}\sum_{i=1}^{n}(\tau)| \tag{20}$$

where $\hat{\tau}_i$ and $\tau_i$ stand for the predicted ITE and the ground truth ITE for the $i$-th instance respectively.

**ACIC 2016.** This is a common benchmark dataset introduced by Dorie et al. (2019), which was developed for the 2016 Atlantic Causal Inference Conference competition data Dorie et al. (2019). It comprises 4,802 units (28% treated, 72% control) and 82 covariates measuring aspects of the linked birth and infant death data (LBIDD). The dataset are generated randomly according to the data generating process setting. We conduct experiments over randomly picked 100 realizations with 63/27/10 train/validation/test splits.

**IHDP.** Hill (2011) introduced a semi-synthetic dataset for causal effect estimation. The dataset was based on the Infant Health and Development Program (IHDP), in which the covariates were generated by a randomized experiment investigating the effect of home visits by specialists on future cognitive scores. it consists of 747 units(19% treated, 81% control ) and 25 covariates measuring the children and their mothers. Following the common settings in Qin et al. (2021); Shalit et al. (2017), We average over 1000 replications of the outcomes with 63/27/10 train/validation/test splits.

**Data Simulation.** In order to verify the effectiveness of our framework in unbiased data, we adopt the generation process proposed in Assaad et al. (2021); Louizos et al. (2017) to simulate the treatment

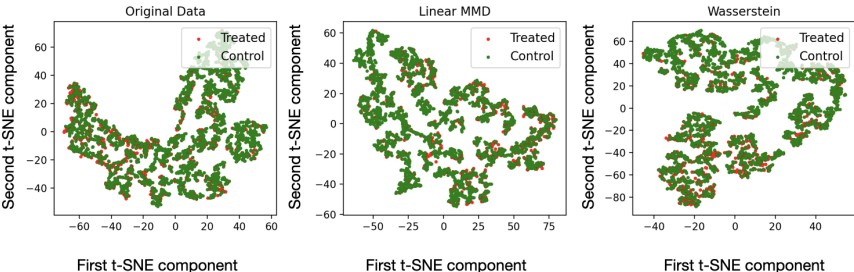

Figure 3: t-SNE visualizations of the balanced representations of ACIC learned by our algorithms $\text{RITE}_{MMD}$ and $\text{RITE}_{WASS}$

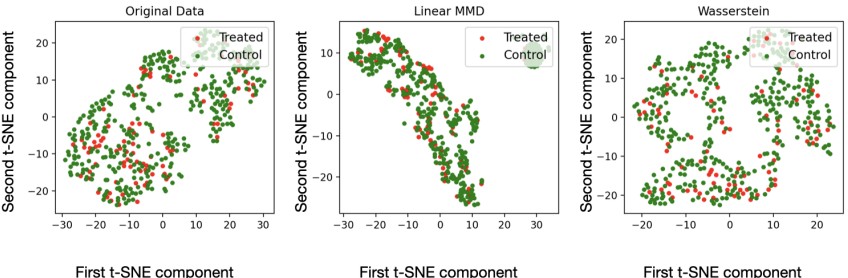

Figure 4: t-SNE visualizations of the balanced representations of IHDP learned by our algorithms $\text{RITE}_{MMD}$ and $\text{RITE}_{WASS}$
effect as:

$$
\begin{aligned}
&\mathbf{x}_i \sim \mathcal{N}(\mu_X, \sigma_X^2); \ \ \mathbf{t}_i | \mathbf{x}_i \sim \text{Bernoulli}(\sigma(\mathbf{x}_i^T \beta_T)) \\
&\epsilon_i \sim \mathcal{N}(0, \sigma_Y^2); \ \ \mathbf{y}_i(0) = \mathbf{x}_i^T \beta_0 + \epsilon_i \\
&\mathbf{y}_i(1) = \mathbf{x}_i^T \beta_0 + \mathbf{x}_i^T \beta_1 + \theta + \epsilon_i
\end{aligned}
\tag{21}
$$

where $\sigma$ is the logistic sigmoid function. This generation process satisfies the assumptions of ignorability and positivity. We randomly construct 100 replications of such datasets with 10,000 units (50% treated, 50% control) and 50 covariates by setting $\sigma_X$ and $\sigma_Y$ both to 0.5, $\beta_T$, $\beta_0$ and $\beta_1$ are sampled from $\mathcal{N}(0, 1)$.

### A.6 LEARNED REPRESENTATIONS

In order to provide more intuitive understandings on the provided explanations of the learned representations between treated and control groups, in this subsection, we conduct the representations studies, where one replication are randomly picked from ACIC and IHDP, respectively, and all of hyperparameters are remained in the default state. We compare the learned representations generated by minimizing our main objective with $\alpha = 0$ and $\beta = 0$, $\text{RITE}_{WASS}$ and $\text{RITE}_{MMD}$ method in terms of $\sqrt{\epsilon_{PEHE}}$ and $\epsilon_{ATE}$ . From the results shown in Figure 3 and 4, we can see: compared to the original data distribution, both $\text{RITE}_{WASS}$ and $\text{RITE}_{MMD}$ can perform several regions where the representations are indeed balanced, so that they appear equal in high-dimension space. Furthermore, between $\text{RITE}_{WASS}$ and $\text{RITE}_{MMD}$ methods, we can find that some regions illustrated in WASS distributions appear a strip-like representation, whereas the linear MMD give rise to a rod-like shape in regions where overlap is small.

### A.7 PSEUDO-CODE OF RITE

The complete algorithm and detailed information about datasets are presented in Algorithm 1 and Table 3.

---

**Algorithm 1:** Learning algorithm of our model

---

1  Indicate the observational data $(x_1, t_1, y_1), ..., (x_m, t_m, y_m)$.
2  Indicate the scaling parameter $\alpha$ and $\beta$ .
3  Initialize all the model parameters.
4  Indicate the epoch number $E$.
5  Compute $u = \frac{1}{m} \sum_{i=1}^{m} t_i$.
6  Compute $w_i = \frac{t_i}{2u} + \frac{1-t_i}{2(1-u)}$ for $i = 1, ..., m$
7  **for** *e in [0, E]* **do**
8      Sample mini-batch data $\mathcal{B}$ from $\mathcal{D}$
9      Compute the gradients of the regularization:

$$g_1 = \nabla_W \beta \mathcal{R}(f)$$

10     Compute the gradients of the IPM term:

$$g_2 = \nabla_W \alpha IPM_G(\hat{p}_\Phi^{t=1}, \hat{p}_\Phi^{t=0})$$

11     Compute the gradients of the empirical loss:

$$g_3 = \nabla_W \frac{1}{|\mathcal{B}|} \sum_{i=1}^{|\mathcal{B}|} w_i L(y_i, f(\Phi(x_i), t_i))$$

12     Obtain the step size scalar $\eta$ with the Adam
13     Update the parameters:

$$W \leftarrow W - \eta(g_1 + g_2 + g_3)$$

14 **end**

---

Table 3: Statistics of the datasets used in our experiments.

| Dataset | #Replications | #Units | #Covariates | Treated Ratio | Control Ratio |
|---------|---------------|--------|-------------|---------------|---------------|
| ACIC | 100 | 4,802 | 82 | 28% | 72% |
| IHDP | 1,000 | 747 | 25 | 19% | 81% |
| Sim | 100 | 10,000 | 50 | 50% | 50% |

