# OpenReview forum: "Bounding the Robustness and Generalization for Individual Treatment Effect"
_ICLR.cc/2024/Conference — ICLR 2024 Conference Withdrawn Submission_

### Official Review · Reviewer_nExb · 2023-10-31

**Soundness:** 3 good
**Presentation:** 3 good
**Contribution:** 2 fair
**Rating:** 5
**Confidence:** 3

**Summary:**

In this paper, the authors point out the problem of some previous works (e.g., CFR), which use the IPM metric to balance representations to reduce confounding bias in CATE estimation, that “transforming IPM to MMD or WASS distances metrics without considering the Lipschitz constant of f and $\Phi$ in the estimation of ITE” results in poor generalization ability and robustness. To address the above problem, the authors give new bounds and claim that constraining the Lipschitz constant term is the key to obtaining robust ITE estimation. They further propose two types of regularizations (respectively for WASS and MMD) called Lipschitz Regularization and reproducing kernel Hilbert space (RKHS) Regularization to encourage robustness in estimating ITE from observational data.

**Strengths:**

1. They address one of the main issues for some previous works (e.g., CFR) of CATE estimation to be applied in real-world applications. That is, the model needs to be more robust and is challenging to optimize with MMD or WASS distance metrics.
2. Their proposed regularization methods are easy to implement, and the source codes are given.

**Weaknesses:**

1. This paper is an improvement of some previous works (e.g., CFR), which use the IPM metric to balance representations to reduce confounding bias in CATE estimation. Though the proposed methods indeed address one of the main issues of these works, they can hardly be applied to most CATE estimation methods, resulting in a relatively limited contribution.
2. In fact, the key challenge for the CFR series methods to be applied in real-world applications is that, due to a lack of counterfactual samples, it is always hard to select appropriate hyper-parameters of the MMD or WASS distance metric. Although the proposed methods use a regularization term to make these methods more robust, the choice of these hyper-parameters is still difficult in real-world applications.
3. In Table 1, I notice that the performance of $\mathrm{RITE_{MMD}}$ is far better than other baselines, including $\mathrm{RITE_{WASS}}$. For example, the mean and std of PEHEs of other methods are all greater than 1, but those of $\mathrm{RITE_{MMD}}$ are 0.45 and 0.06, respectively. Can the authors please make an explanation for this unusual observation?
4. There are some clerical errors, including mathematical symbols, that need to be clarified.

**Questions:**

See weakness

---

### Official Review · Reviewer_UoGx · 2023-10-31

**Soundness:** 1 poor
**Presentation:** 2 fair
**Contribution:** 3 good
**Rating:** 5
**Confidence:** 4

**Summary:**

The paper deals with robustness of Individual Treatment Effect (ITE) estimation based on neural networks, with respect to small perturbation to the covariates.

Assuming Lipschitz-continuity on the neural network and an $||\cdot||_p$ loss, they provide a theoretical upper bound for the expected ITE estimation error. They derived that this upper bound depends on the network's Lipschitz constant, as well as the empirical regression losses on the training set, the covering number of the observational data distribution and distance between treatment and control distribution.

They introduce two new algorithms for ITE estimation based on these insights. Both are aiming to reduce the Lipschitz constant of the neural network during training and therefore, impose specific type of regularisation. One algorithm, called RITE_WASS, is based on the Wasserstein distance and imposed orthonormality in the weight matrices of the transformation layer. The other,  referred to as RITE_MMD,  is based on the maximum mean discrepancy and uses some reproducing kernel Hilbert space regularisation.

They perform experiments on synthetic and semi-synthetic datasets, as well as experiments in which they artificially add noise to the covariates.

**Strengths:**

*Overall*
1. The paper is original. It seems to be the first attempt to apply ideas of adversarial machine learning to ITE estimation. This might not only help against malicious attempts to influence ITE prediction, but also against small measurement errors inherent in many datasets collected for ITE estimation.

*Methods and experiments*

2. The proposed methods are theoretically founded.
3. The paper provides an extensive comparison of their proposed methods to other state-of-the-art ITE estimators. The point estimate of their method's performance is lower than the one of all other estimators in all settings, and significantly lower than all other methods in the added noise setting.

**Weaknesses:**

*Methods and experiments*

1. The performance of their methods on the dataset without added noise is not significantly different from the performance of other state-of-the-art ITE estimator based on neural networks, e.g. TARNet. Hence, it does not convincingly outperform the other methods in a standard setting.
2. In the experiment with added noise, the noise for each feature is $U(-1,1)$ distributed. An analysis for a shift of the performance for different noise levels is missing.

*Theory*

3. The neural net is introduced with the notation $f$. Based on that the Lipschitz-constant $\Lambda_p$ is defined. In the theory part, the variable name $f$ is assigned to something else, namely the hypothesis of predicting the outcome given the representative covariates $\Phi(x)$ and the treatment assignment $t$. This is confusing, especially when the Lipschitz property is applied to f(\Phi(x),t), e.g. as in the proof of Theorem 1.

4. In the proof of Theorem 2 part of the reasoning is unclear to me:

- After the covering number for the data distributions and $\gamma$ balls is introduced, the proof goes on to partition the support of the data distribution into a number of subsets depending on the  $\gamma/2 $ covering number of $\mathcal{X}$ and $\mathcal{Y}$. A proper reasoning why this partition is possible is missing. (Moreover, I didn't find a formal introduction of $\mathcal{X}$ and $\mathcal{Y}$. It seems implicitly assumed that they represent the support of X and Y, respectively.)

- A new variable $\epsilon$ is introduced in the last equation, which isn't defined anywhere in the theorem nor the proof.

- The last inequality seems overall questionable to me. In the first term the loss function takes in the arguments $(y_j, f(\Phi(x_j),t_j)$ and $(y, f(\Phi(x),t)$ for some arbitrary $(y,x,t)\in\mathcal{D_i}$ that maximise the different between the output of the loss function. In general, I would assume that $(y_j, x_j,t_j) \neq (y, x,t)$ in each coordinate.
     - The constant $\lambda_p$ is defined as the Lipschitz-constant of the loss $L(\cdot,\cdot)$ with respect to the second argument. Therefore, the difference of the loss could not be bounded by this Lipschitz-constant for $y_j \neq y$.
     - The network function $f(\cdot,t_j)$ and $f(\cdot,t)$ may differ for $t_j\neq t$. The Lipschitz-constant $\Lambda_p$ is defined for one specific neural network, so either $f(\cdot,t_j)$ or $f(\cdot,t)$. But it is not defined for the difference between both of them, at least in my understanding.

5. The definition 7 gives an alternative definition/bound for the Lipschitz constant of a neural network. It is not clear how this relates to the Lipschitz constant in (4).
6. The definitions and proofs of the paper are not very detailed.

**Questions:**

*Methods and experiments*
1. It would be quite interesting to see the performance of the representative deep learning methods for different noise levels, preferably in a plot with confidence bounds around performance estimates.
2. Do you have an intuition why the method based on Wasserstein is "more suitable for balancing the representation"? What are possibly characteristics of a dataset that make one or the other metric more suitable?

*Theory*

3. Could you please send a corrected/detailed version of the proof of Theorem 2? As it proves the main argument for the methods, namely bounding using Lipschitz-constants,  I am particular interested in that.

---

### Official Review · Reviewer_FvrM · 2023-11-01

**Soundness:** 2 fair
**Presentation:** 2 fair
**Contribution:** 2 fair
**Rating:** 3
**Confidence:** 3

**Summary:**

While the paper claims individual treatment effect (ITE) estimation, this paper actually studies parametric conditional average treatment effect (CATE) estimation when unconfoundedness and SUTVA assumption holds (which is not ITE estimation, which is estimating Y_i(1)-Y_i(0).) Specifically, they apply typical approaches to bound the imputation bias (or generalization bias) using popular theoretical tools in neural network analysis. Motivated by the theoretical bounds, they propose a new method called RITE, which empirically performs quite well relative to other baseline CATE estimation frameworks in terms of some popular datasets.

**Strengths:**

I am not an expert in empirical methods for CATE estimation. If I can assume that their choice of datasets and baselines are flawless, their proposed method (RITE), which is motivated by theoretical ideas, is quite appealing. Their proposed method exceeds other baseline methods a lot. Assuming that they are going to reformat the entire structure of their paper to claim their contribution to CATE estimation world instead of ITE estimation world, their result looks interesting.

**Weaknesses:**

As I described in the Summary, their definition of ITE is actually CATE, with additional assumptions. However, this paper is framing itself to be the first one to propose a robustness bound in the ITE estimation world, which makes me hard to vote for acceptance.
For the paper which indeed discusses ITE estimation, see “Conformal inference of counterfactuals and individual treatment effects”, Lei and Candes, 2021.

**Questions:**

I am curious if you would be willing to reframe the contribution of this paper in terms of CATE estimation. I think in general the paper includes good and interesting contributions in terms of CATE estimation, which is actually shown well in their experiment comparing their method with other CATE estimation methods.

---

### Official Review · Reviewer_WUez · 2023-11-05

**Soundness:** 2 fair
**Presentation:** 2 fair
**Contribution:** 1 poor
**Rating:** 5
**Confidence:** 3

**Summary:**

The paper focuses on the problem of robustness of representation learning methods for conditional average treatment effect estimation (CATE) estimation. Specifically, the authors consider robustness in the sense of adversarial examples, which might occur in medical practice. The authors extend the CATE generalization bounds from Shalit et al. (2017) so that they depend on the Lipschitz constant of the model. Then, by controlling the Lipschitz constant of the model, we can guarantee adversarial robustness. In practice, this can be achieved with two types of regularisations. Ultimately, the paper demonstrates the effectiveness of those regularisations on the standard CATE estimation benchmarks and several ablation studies. Additionally, to verify the robustness capabilities, the authors provide a study with noise-corrupted test data.

**Strengths:**

Adversarial robustness is very relevant for the reliability and widespread application of CATE representation learning methods. Thus, I appreciate the author’s effort to extend a theory for CATE generalization bounds from Shalit et al. (2017). Specifically, the paper provided theoretic and empirical results on how a Lipschitz constant of a model affects its generalisation performance.

**Weaknesses:**

While this paper studies an important problem, there are several issues:
1. Contrasting the previous work. A large part of the derivations and theoretic results seem to mirror the work of Shalit et al. (2017). I would encourage authors to better highlight the novelty of their contribution, e.g., they could compare the bounds provided by Shalit et al. (2017) or Johansson et al. (2022) with the results of Theorems 2 and 3. The later work, namely, Johansson et al. (2022), is not mentioned in the related work, although it is very relevant to the theory.
2. Fair comparison. I am concerned, that a comparison with other baselines might be unfair. First, I do not understand, why DragonNet is used as a baseline, as it aims at estimating ATE. Additionally, QHTE is used as a baseline, even though it should be used for budgeted CATE learning (which is a different task). Second, the original methods, i.e., TARNet or CFR are supposed to have some regularisation in them, like L1/L2 regularisation. Enforcing adversarial robustness could also be seen as a kind of regularisation. Therefore, I would expect TARNet and CFR baselines from Tables 1 and 2 to contain some simple regularisation, e.g., L1/L2. Otherwise, the whole performance gain of the RITE simply comes from using over-parametrized models (e.g., the authors used 3 hidden layer MLPs) on small benchmark datasets. Additionally, to make the experimental results fully transparent, I encourage authors to report a hyper-parameter tuning procedure in more detail.

Also, I have some minor concerns. For example, the citation format seems to be wrong, the syntax of formulas is sometimes wrong, and the enumeration of the theorems in the Appendix is misleading. Some terms are also not defined throughout the paper, e.g., $\epsilon$ in Theorem 1, $d$ in $C_d$

I am happy to raise my score if the authors address all my concerns during the rebuttal.

References:
- Uri Shalit, Fredrik D Johansson, and David Sontag. Estimating individual treatment effect: generalization bounds and algorithms. In International Conference on Machine Learning, pp. 3076–3085. PMLR, 2017.
- Johansson, F. D., Shalit, U., Kallus, N., & Sontag, D. (2022). Generalization bounds and representation learning for estimation of potential outcomes and causal effects. The Journal of Machine Learning Research, 23(1), 7489-7538.

**Questions:**

- Does the Lipschitz constant in Theorem 1 refer to the whole model, or the model without the representation layers?
- Is Theorem 4 really necessary? It looks like a copy of Theorem 3.
- Shouldn’t the Lipschitz constant for the loss, i.e., $\lambda_p$ be 1 when $G$ is a family of 1-Lipshitz functions?